# Monte Carlo Deep Neural Network Arithmetic

## Abstract

Quantization is a crucial technique for achieving low-power, low latency and high throughput hardware implementations of Deep Neural Networks. Quantized floating point representations have received recent interest due to their hardware efficiency benefits and ability to represent a higher dynamic range than fixed point representations, leading to improvements in accuracy. We present a novel technique, Monte Carlo Deep Neural Network Arithmetic (MCDA), for determining the sensitivity of Deep Neural Networks to quantization in floating point arithmetic. We do this by applying Monte Carlo Arithmetic to the inference computation and analyzing the relative standard deviation of the neural network loss. The method makes no assumptions regarding the underlying parameter distributions. We evaluate our method on pre-trained image classification models on the CIFAR-10 and ImageNet datasets. For the same network topology and dataset, we demonstrate the ability to gain the equivalent of bits of precision by simply choosing weight parameter sets which demonstrate a lower loss of significance from the Monte Carlo trials. Additionally, we can apply MCDA to compare the sensitivity of different network topologies to quantization effects.[1]

## 1 Introduction

Deep Neural Networks have achieved state-of-the-art performances in many machine learning tasks such as such as speech recognition (Collobert et al., 2011), machine translation (Bahdanau et al., 2014), object detection (Ren et al., 2015) and image classification (Krizhevsky et al., 2012). However, excellent performance comes at the cost of significantly high computational and memory complexity, typically requiring teraops of computation during inference and Gigabytes of storage. To overcome these complexities, compression methods have been utilized, aiming to exploit the inherent resilience of DNNs to noise. These engender representations which maintain algorithm performance but significantly improve latency, throughput and power consumption of hardware implementations. In particular, exploiting reduced numerical precision for data representations through quantization has been emphatically promising, whereby on customizable hardware, efficiency scales quadratically with each bit of precision.

Quantization of fixed-point arithmetic (Q-FX) for DNN inference has been extensively studied, and more recently there has been increasing interest in quantized floating point (Q-FP) arithmetic for both DNN inference and training (Wang et al., 2018). Q-FP has the advantage of higher dynamic range compared to equivalent Q-FX representations and reduced hardware cost over single-precision floating point (FP). This has influenced application specific integrated circuits (ASICs) such as Google's tensor processing unit (TPU), which supports 16-bit floating point (16-FP) and soft processors such as Microsoft's Project Brainwave which utilizes 8-FP.

To illustrate these hardware benefits, we synthesized arithmetic logic units (ALUs) in different formats and different precision on an FPGA and present performance estimates in operations per second (OPs) and area estimates in Look-up Tables (LUTs) per operation (LUTs/Op) in Figure 1. As shown, 8-bit fixed point (8-FX) achieves improved performance and area over 8-FP. However, 7-FP is a significant improvement over 8 or 12-FX and 8 or 9-FP. These examples demonstrate substantial performance and area benefits from reducing FP precision by only 1 to 2-bits. Thus, if we

---

[1]Source code will be available if the paper is accepted

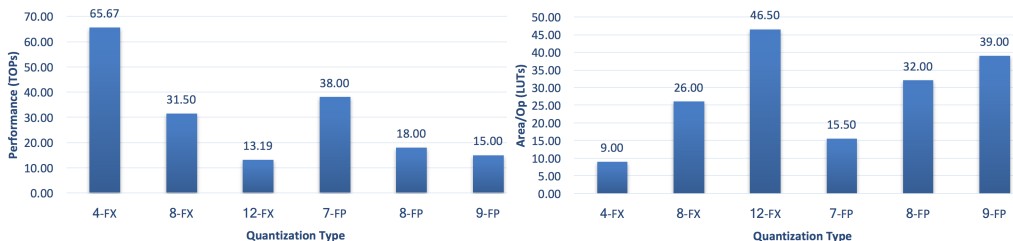

Figure 1: The estimated multiply-accumulate performance on a Xilinx VU9P FPGA, operating at 500MHz, in teraops (left) and area (right) for fixed (FX) and floating-point (FP) in various precisions (both weights and activations).

can design networks which not only achieve high accuracy but are robust to quantization, higher performing hardware solutions are possible.

Since in Q-FP, we are trying to represent the infinite set of real numbers using a finite number of bits, quantization and rounding artefacts will be introduced, with inaccuracies being cascaded along the computation graph (IEEE, 1985; Goldberg, 1991; Higham, 2002). This paper proposes Monte Carlo Deep Neural Network Arithmetic (MCDA), a novel way to apply Monte Carlo Arithmetic (MCA) (Parker et al., 2000) for determining the sensitivity of Deep Neural Networks to Q-FP representations. It allows hardware-software designers to quantify the impact of quantization, enabling more efficient systems to be discovered. We do this by exploiting Monte Carlo simulations under which rounding effects are randomized. This, in turn, allows one to infer the sensitivity of executing a computation graph to quantization effects.

Our MCDA technique is highly sensitive, allowing very small differences in quantization behaviour to be detected. The technique makes no assumptions regarding data distributions and directly measures the effects of quantization on the problem under study. This allows us to provide insights into the precision requirements of any inference network for any given dataset. Additionally we can use the technique to select weight parameters which are more robust to floating point rounding. The theoretical and practical contributions of this work can be summarized as follows:

- We introduce a novel and rigorous analysis technique, Monte Carlo Deep Neural Network Arithmetic (MCDA), which measures the sensitivity of Deep Neural Network inference computation to floating point rounding error.
- When applied to neural network inference, we show MCDA can determine the precision requirements of different networks, rank them, and detect small differences between different neural network topologies and weight sets.
- We demonstrate that while a network with the same topology but different weights may have the same loss and validation accuracy, their sensitivity to quantization can be vastly different. Using the CIFAR-10 and ImageNet datasets, we introduce a method to choose weights which are more robust to rounding error, resulting in a greatly improved accuracy-area tradeoff over state-of-the-art methods.

It is worth noting that although we consider convolutional neural networks for image classification, this method could be applied for any neural network model architectures and applications. Moreover, while the experiments in this paper are limited to inference, it may be possible to apply the same idea to analyze training algorithms.

## 2 RELATED WORK

Low-precision representations of deep learning have been extensively studied. Many training methods have been developed to design representations for fixed point inference (Jacob et al., 2018; Faraone et al., 2018; Zhou et al., 2016) and training (Wu et al., 2018b; Yang et al., 2019; Gupta et al., 2015; Sakr & Shanbhag, 2019). Other methods have also utilized Q-FP arithmetic for inference and training whilst maintaining single-precision accuracy. (Micikevicius et al., 2018) implemented 16-FP arithmetic training whilst storing a 32-FP master copy for the weight updates. Additionally,

(Wang et al., 2018; Mellempudi et al., 2019) with 8-FP arithmetic whilst using a 16-FP copy of the weights and 16/32-bits for the accumulator. Techniques for determining per-layer sensitivity to quantization have also been studied (Choi et al., 2016; Sakr & Shanbhag, 2018). Further, other studies have successfully determined the minimum fixed point precision requirements for a given DNN accuracy threshold (Sakr et al., 2017). The accuracy and stability of various numerical algorithms in finite precision arithmetic has been studied in (Higham, 2002; Wilkinson, 1994). This has led to techniques for tracking information lost from finite precision arithmetic using random perturbation such as Monte Carlo Arithmetic (Parker et al., 2000; Frechtling & Leong, 2015). Monte Carlo methods have also been used in Bayesian Neural Networks (Buchholz et al., 2018; Blier & Ollivier, 2018). In particular, (Achterhold et al., 2018) introduced a quantizing prior to learn weights which are either close to a quantized representation or have high variance. (Louizos et al., 2017) used hierarchical priors to prune nodes and posterior uncertainties to determine the optimal fixed point precision. (Blundell et al., 2015) use a Monte Carlo approach to learn a probability distribution on the weights of a neural network. To the best of our knowledge, our work is the first to present a technique for directly determining the sensitivity of DNNs to floating point rounding and to explicitly compute precision bounds of a trained network. These ideas can be very usefully applied to extending the limits of low-precision representations in deep learning applications.

## 3 BACKGROUND

In this section, we describe background theory upon which our technique for determining the sensitivity of DNNs to floating point rounding is based.

### 3.1 FLOATING POINT ARITHMETIC

The IEEE-754 binary floating point format (IEEE, 1985) represents most real numbers $x$ by a subset in normal form as:

$$\hat{x} = (-1)^{s_x}(1 + m_x)2^{e_x} \tag{1}$$

where $s_x \in \{0, 1\}$ is the sign bit, $e_x$ is an integer representing the exponent of $\hat{x}$ and $m_x$ is the mantissa of $\hat{x}$. Such number formats can be described as a $(s_x, e_x, m_x)$ tuple. In binary form the representation is $(b^s, b_1^e, b_2^e, ..., b_{B_{e_x}}^e, b_1^m, b_2^m ..., b_{B_{m_x}}^m) \in \{0, 1\}^B$, with $B_{e_x}$ and $B_{m_x}$ being the number of exponent and mantissa bits, respectively. The infinite set of real numbers $\mathbb{R}$ is represented in a computer with $B = 1 + B_{e_x} + B_{m_x}$ bits, and we define the finite set of real numbers representable in floating point format as *exact values*, $\mathbb{F} \subset \mathbb{R}$. Real numbers which aren't representable are rounded to their nearest exact value. We call this set of numbers *inexact values*, $\mathbb{I}$, where $\mathbb{I} \cup \mathbb{F} = \mathbb{R}$.

The approximation $\hat{x} = \mathbb{F}(x) = x(1 + \delta)$, given $x \in \mathbb{I}$, introduces rounding error into the computation. The value of $\delta = \left\| \frac{x - \hat{x}}{x} \right\|$, represents the relative error which is a function of the machine hardware precision, $p$, as $\delta \leq \epsilon$, where $\epsilon = 2^{-p}$ (IEEE, 1985; Goldberg, 1991; Higham, 2002).

In general, inexactness can be caused by finite representations or errors propagating from earlier parts of the computation. Often the primary cause of error in floating point arithmetic is *catastrophic cancellation* which causes numerical inaccuracy. Catastrophic cancellation occurs when for example, two near equal FP numbers, sharing $k$ significant digits, are subtracted from one another as shown in (2) (Higham, 2002).

$$
\begin{array}{ll}
\begin{array}{l}
0. \; f_1 \; f_2 \; ... \; f_k \; f_{(k+1)} \; ... \; f_t \\
- \; 0. \; f_1 \; f_2 \; ... \; f_k \; g_{(k+1)} \; ... \; g_t \\
\hline
= 0.0 \quad 0 \; .... \; 0 \; h_{(k+1)} \; ... \; h_t
\end{array}
& (2)
\end{array}
\qquad
\begin{array}{ll}
\begin{array}{l}
0. \; f_1 \; f_2 \; ... \; f_k \; f_{(k+1)} \; ... \; f_t \; r_{(t+1)} \; ... \; r_p \\
- \; 0. \; f_1 \; f_2 \; ... \; f_k \; g_{(k+1)} \; ... \; g_t \; \hat{r}_{(t+1)} \; ... \; \hat{r}_p \\
\hline
= 0.0 \quad 0 \; .... \; 0 \; h_{(k+1)} \; ... \; h_t \; i_{(t+1)} \; ... \; i_p
\end{array}
& (3)
\end{array}
$$

In normalized form, the leading zeros are removed by shifting the result to the left and adjusting the exponent accordingly. The result is $0.h_{k+1}...h_t i_1...i_k$ which has only $(t - k)$ accurate digits and digits $i$ which are unknown. Additionally, the remaining accurate digits $h$ are most likely affected by rounding error in previous computations. This can significantly magnify errors, especially in computing large computational graphs such as that of state-of-the-art DNNs.

If either operand in (2) is inexact, then the digits $h$ are no more significant than any other sequence of digits. Yet, FP arithmetic has no mechanism of recording this loss of significance. By padding both our operands with random digits $r$ and $\hat{r}$ in (3), the resulting digits $i$ are randomized. If $k$

digits are lost in the result, then $k$ random digits will be in the normalized result and when computed over many random trials, the results will disagree on the trailing $k$ digits. In this case, we are able to detect catastrophic cancellation because the randomization over many trials provides a statistical simulation of round-off errors. We can use techniques from numerical analysis such as Monte Carlo methods to appropriately insert precision-dependant randomization in this way.

### 3.2 MONTE CARLO ARITHMETIC

Monte Carlo methods can be used to analyze rounding by representing inexact values as random variables (Parker et al., 2000; Frechtling & Leong, 2015). The real value $x$, as represented in (1), can be modelled to t digits, using:

$$inexact(\hat{x}, t, \delta) = \hat{x} + 2^{e_x - t}\delta = (-1)^{s_x}(1 + m_x + 2^{-t}\delta)2^{e_x} \quad (4)$$

where $\delta \in U(-\frac{1}{2}, \frac{1}{2})$ is a uniformly distributed random variable and $t$ is a positive integer representing the *virtual precision* of concern. For the same input $\hat{x}$ in (4), we can run many Monte Carlo trials which will yield different values on each trial, where $0 < t < p$ so that the MCA can be run accurately on a computer with machine precision $p$. The ability to vary $t$ is useful because it allows us to then evaluate the hardware precision requirements of a given system or computational graph for a given DNN.

MCA is a method to model the effect of rounding on a computational graph by randomizing all arithmetic operations. The randomization is applied for both generating inexact operands and also in rounding. In each operation using MCA, ideally both catastrophic cancellation and rounding error can be detected. An operation using MCA is defined as:

$$x \circ y = round(inexact(inexact(x) \circ inexact(y))) \quad (5)$$

where $\circ \in (+, -, \times, \div)$. By applying the inexact function to both operators we make it possible to detect catastrophic cancellation. Furthermore, applying the inexact function to the operation output and then rounding this value implements random rounding and hence is used to detect rounding error (Parker & Langley, 1997). Hence, for the same input into the system, each trial will yield different operands and output.

After modifying the inexact and rounding operations as described, we use random sampling to simulate Monte Carlo trials. For each trial, we collect data on the resulting output of the system and compute summary statistics to quantify its behaviour (Parker et al., 2000). With sufficiently large number of Monte Carlo trials and virtual precision $t$, the expected value of the output from these trials will equal the value from using real arithmetic. As explained in the next section, we can determine the total number of digits lost to rounding error and the minimum precision required to avoid a total loss of significance.

### 3.3 ANALYSIS

The relative error is bounded by $\delta \leq 2^{-p}$ from the design of IEEE FP arithmetic (Wilkinson, 1994; Goldberg, 1991). With this inequality, we can determine the expected number of significant binary digits available from a p-digit FP system as $p \leq -log_2(\delta)$. These definitions can be adapted for MCA by replacing the precision of the FP system, $p$, by the virtual precision, $t$, of an MCA operation. Thus, the relative error of an MCA operation, for virtual precision $t$, is $\delta \leq 2^{-t}$ and the expected number of significant binary digits in a $t$-digit MCA operation is at most $t$. Using this definition and the proof provided in (Parker & Langley, 1997), the total significant binary digits in a set of $M$ trials is $s' = log_2\frac{\mu}{\sigma}$ where $\mu$ is the mean and $\sigma$ the standard deviation. The output of the system should be some scalar value so that we can perform such analysis. For experimentation, $M$ trials are run for all of $t \in \{1, 2, 3, ..., t_{max}\}$. The total number of base-2 significant digits lost in a set of $M$ trials is $K_t$, in (6):

$$K_t = t - s' = t - \log_2(\frac{\mu}{\sigma}) = \log_2 \Theta + t \quad (6)$$

where $\Theta = \frac{\sigma}{\mu}$, for ($\mu \neq 0$), is the relative standard deviation (RSD) of the MCA results. The virtual precision $t$ controls the perturbation strength applied by the inexact function. For a given $K_t$, as we reduce $t$, the RSD should increase according to equation 6. At some point, an unexpected loss of

significance (Frechtling & Leong, 2015) is encountered due to the nonlinear effects of quantization. The value at which this occurs is defined as $t_{min}$. The number of significant digits lost for the system being analyzed is then computed by averaging all $K_t$ whereby $t > t_{min}$ as shown in (7):

$$K = \begin{cases} \frac{1}{t_{max} - t_{min}} \sum_{t=t_{min}}^{t_{max}} K_t & where \quad t_{max} > 1 \\ K_t & where \quad t_{max} = 1 \end{cases} \tag{7}$$

For DNN inference, we propose to use $K$ as a sensitivity measure for the network to FP rounding. The method for implementing this is discussed in the next section.

## 4 MONTE CARLO DEEP NEURAL NETWORK ARITHMETIC

We now describe MCDA, a methodology for applying MCA techniques to DNN computation, allowing us to understand the sensitivity of a given network and its weight representation to FP rounding.

### 4.1 NETWORK MODEL

We consider a generalized non-linear $L$-layer neural network with an output vector $\mathbf{y}_L$, input data vector $\mathbf{x}$ and learnable weight parameter tensor $\mathbf{w} = \bigcup \mathbf{w}_l$ ($l = 1, \ldots, L$), whereby $\mathbf{y}_L = f(\mathbf{x}; \mathbf{w})$. To compute $\mathbf{y}$, several layers consisting of general matrix multiplication (GEMM) operations (such as convolutional and fully-connected layers) between the layer input $\mathbf{x_l}$ and weight parameters $\mathbf{w}_l$, followed by a non-linear activation function, $h$, producing intermediate layer outputs $\mathbf{x_l}$, i.e. $\mathbf{y}_l = h(\mathbf{x_l} \otimes \mathbf{w_l})$. The output of a given layer becomes the input to the subsequent layer, i.e. $\mathbf{x}_{l+1} = \mathbf{y}_l$, with $\mathbf{x}_1 = \mathbf{x}$. A loss function is the objective function to minimize updating $\mathbf{w}$ via an optimizer such as stochastic gradient descent. For a given set of input data $X$, the total network loss during inference is calculated by applying a loss function $loss(f(\mathbf{x}; \mathbf{w}), \hat{\mathbf{y}}(\mathbf{x}))$ where $\hat{\mathbf{y}}(\mathbf{x})$ is the target ground truth output for $\mathbf{x}$. The total loss for $X$ is then a scalar output, such that:

$$L(X; \mathbf{w}) = \frac{1}{|X|} \sum_{\mathbf{x} \in X} loss(f(\mathbf{x}; \mathbf{w}), \hat{\mathbf{y}}(\mathbf{x})) \tag{8}$$

Updates of $w$ are usually done in small batches over subsets of $X$.

Naively applying MCA to each operation (fine-grained MCA) as described in equation (5) poses significant computational difficulties for DNN models. We observe two primary issues with employing fine-grained MCA to a DNN computational graph:

- Firstly, the number of required trials for Monte Carlo experiments to generate robust results can typically be in the hundreds or thousands. As DNN inference of state-of-the-art networks typically consist of billions of operations, the computational requirements of applying MCA after each operation will be very large, making the technique impractical.

- Using the accuracy as the system output for MCA experiments is problematic because it is a discrete value. For high values of $t$, Monte Carlo results across different $t$ then become indistinguishable and the standard deviation for a given $t$ is potentially zero.

### 4.2 MONTE CARLO NETWORK INFERENCE

To reduce the computational cost of Monte Carlo experiments, we employ MCDA, which is a coarse-grained approach to MCA for GEMM operations. Conveniently, these can be naturally implemented in modern machine learning frameworks such as PyTorch. Furthermore, to ensure the system output is a continuous value, the loss function output is used, rather than the accuracy. In this case, small perturbations in layer operands are more likely to produce observable changes in the output.

MCDA applies a vector version of (5) to the DNN inference computational problem in (8). Our operands in this case are vectors and $\circ$ represents a neural network layer operation. For example, the output from performing a GEMM operation can thus be represented by:

$$\mathbf{y_l} = round(inexact(inexact(\mathbf{x_l}) \otimes inexact(\mathbf{w_l}))) \tag{9}$$

Since the inexact function is applied to the inputs and outputs of a GEMM operation, an optimized implementation can be used. This is in contrast to full MCA which requires the application of (5)

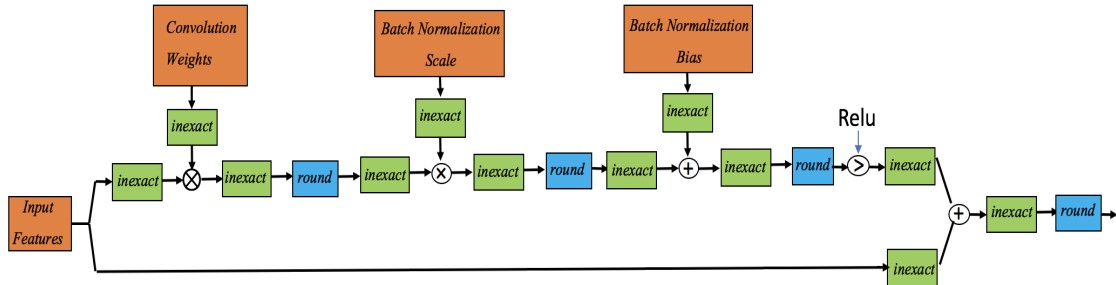

Figure 2: Applying MCDA to residual blocks found in ResNet. The inexact function is applied to each input operand and both the inexact function and base-2 rounding is applied to the output.

to every individual scalar operation. The vector form in (9) is applied to each edge of neural network computational graph where multiply (division) and/or add (subtract) operations are performed. Hence, it is not applied to operations such as MaxPool and ReLu. As an example, in Figure 2 we show where the inexact function is applied for a residual block with folded batch normalization, which is a repeating sequence of layers found in ResNet models (He et al., 2015). At the final output of the network, the loss is computed with (8) and the inexact function is applied to the outputs $\mathbf{y}$ and also the loss output scalar value $L$. From analyzing the behavior of the loss, we infer the sensitivity of the accuracy of the system to FP rounding. By using MCDA for the GEMM operations, we will not be able to detect all instances of catastrophic cancellation. However, we significantly reduce execution time over fine-grained MCA and show in the next section that we can still retrieve valuable information about our system. In fact, for one trial with one batch of 32 images on ImageNet running on a Nvidia Titan Xp GPU, the speed up of regular inference without MCDA is only $1.05\times$ (for a single Monte Carlo trial). We also note that fine-grained MCA could be applied with a simple modification and would be possible given appropriate customized hardware support for parallel Monte Carlo computations (Yeung et al., 2011).

## 5 EXPERIMENTAL RESULTS

In this section, we present experimental results for applying MCDA to exemplary convolutional neural networks. We use the CIFAR-10 and ImageNet image classification datasets to compare MobileNet-v2 (Sandler et al., 2018), EfficientNet (Tan & Le, 2019), AlexNet, ResNet (He et al., 2015), SqueezeNet (Iandola et al., 2016) and MnasNet (Tan et al., 2018). For CIFAR-10, we use a batch size of 128, whilst we use a batch size of 32 for ImageNet experiments. Cross-entropy is used as the loss function for both datasets. For a given network, dataset, weight representation and $t$, we apply MCDA with $M = 1000$ trials. The resulting loss from each trial is computed with the same single batch of images and hence additional data is not required. We compute $\Theta_t$ from our results for all $t \in \{1, 2, 3, ..., 16\}$. Following this, we run linear regression analysis using the MCALIB[2] (Frechtling & Leong, 2015) R library, on our $\Theta_t$ values, to determine $t_{min}$ and $K$. $t_{min}$ is defined as the point of lowest $t$ where the difference between the regression line and the equivalent $\Theta_t$ is less than half a binary digit, i.e. $\log_{10}(2^{0.5})$. Further detail on the calculation of $K$ and $t_{min}$ from MCALIB can be found in Appendix A.1. We report Q-FP validation accuracy using the quantization function from (Wang et al., 2018) with stochastic rounding[3] (See Appendix A.2).

### 5.1 DISTINGUISHING WEIGHT PARAMETER REPRESENTATIONS

As discussed in Section 3.1, the inexactness in FP arithmetic largely depends on the numerical value of operands. Two instances of the same network and dataset, with the same validation accuracy, but vastly different weight representations, will likely produce differing sensitivities to FP rounding. We first train 8 instances of EfficientNet-b0 and MobileNet-V2 on CIFAR-10 from scratch with random initialization from (Glorot & Bengio, 2010), all achieving within 1% validation accuracy of one another. Using MCDA, we calculate the $K$ values for each model (See Appendix A.3).

---

[2]https://github.com/mfrechtling/mcalib
[3]https://github.com/Tiiiger/QPyTorch

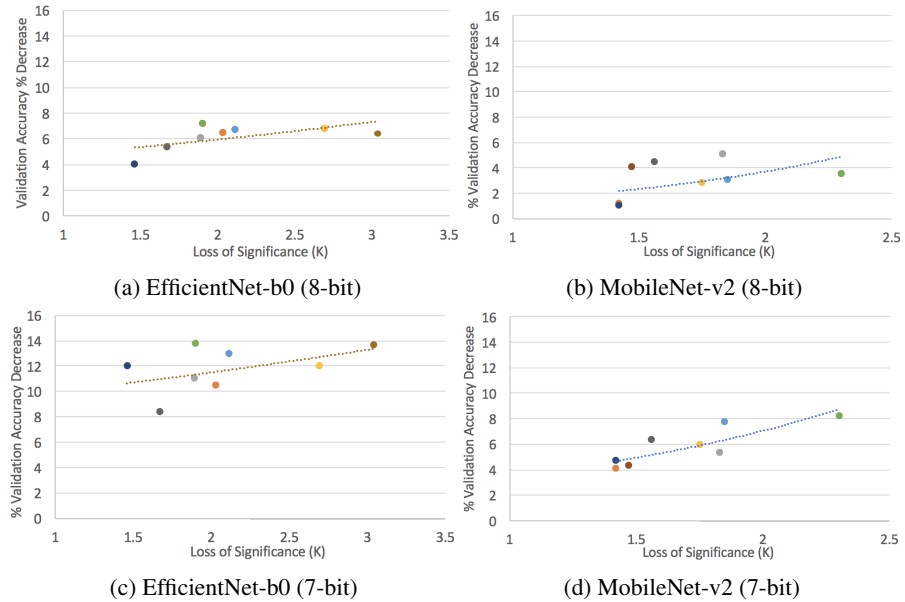

(a) EfficientNet-b0 (8-bit)      (b) MobileNet-v2 (8-bit)

(c) EfficientNet-b0 (7-bit)      (d) MobileNet-v2 (7-bit)

Figure 3: CIFAR-10 percentage validation accuracy decrease from single-precision for 8 different trained models when using post-training Q-FP representations at different precisions.

Table 1: Post-training Q-FP validation accuracy on CIFAR-10 when using $K$ for model selection rather than single-precision validation accuracy

| | MobileNet-v2 | | EfficientNet-b0 | |
|---|---|---|---|---|
| Precision | MCDA | (Wang et al., 2018) | MCDA | (Wang et al., 2018) |
| (Weight, Act.) | ($K =$1.42) | ($K =$1.56) | ($K =$1.46) | ($K =$1.89) |
| (32,32) | 89.6 | **90.3** | 93.7 | **94.6** |
| (8,8) | **88.5** | 86.7 | **90.0** | 89.0 |
| (6,6) | **79.0** | 76.6 | **64.6** | 56.0 |
| (5,5) | **48.4** | 40.8 | **22.0** | 18.6 |

We then test their percentage validation accuracy decrease from using post-training quantization (i.e. no finetuning) with varying Q-FP precisions. In Figure 3, we see that the models with higher $K$ values typically experience a larger drop in Q-FP accuracy, indicating they are more sensitive to floating point rounding error. Notably, the model with lowest $K$ for 7-bit MobileNet-v2 experiences a lower percentage validation accuracy drop than three of the 8-bit models. In this case, MCDA model selection enables the saving of a bit of precision while achieving smaller accuracy decrease than some of the trained 8-FP models.

## 5.2 Comparison To Previous Work

One practical use case from the insights gained by MCDA is model selection for quantization. Typically when quantizing a given model trained on a given dataset, the model with highest validation accuracy is chosen and the sensitivity to quantization is assumed to be the same across models. As discussed, for Q-FP representations this is not necessarily the case. We can use $K$ from MCDA to predict which models will be more robust to quantization. To demonstrate this, in Table 1 we compare post-training quantization results for model selection based on $K$ from MCDA, against a baseline model chosen based on the highest single-precision validation accuracy. Evidently, even though the single-precision accuracy is initially as much as 0.9% higher, after quantizing the network to 8-5 bits, the accuracy of the network chosen by smallest $K$ is always significantly higher.

## 5.3 Network Comparison

Modern DNNs consist of convolutional blocks with highly varying computational graphs (Wu et al., 2018a; Howard et al., 2017). Using MCDA we can also compute and compare their sensitivites to floating point rounding error to determine which networks will be robust to Q-FP representations.

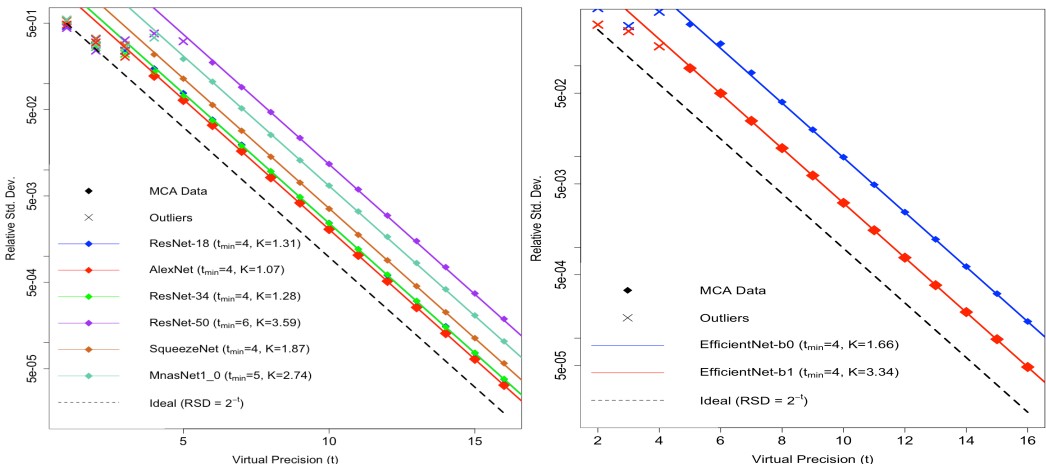

Figure 4: Comparison of $K$ and $t_{min}$ for various networks (Right) and EfficientNet variants (Left) at different virtual precisions on the ImageNet dataset.

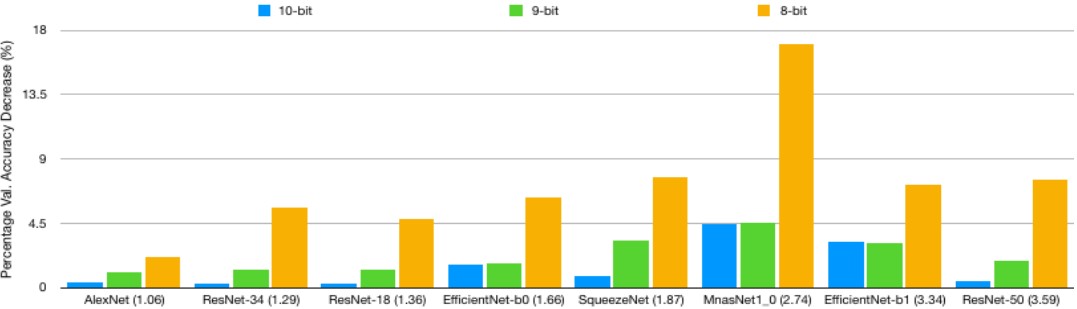

Figure 5: Q-FP percentage validation accuracy decrease for different ImageNet models

In Figure 4 we show the $\Theta_t$ of pre-trained models, trained on the ImageNet dataset from PyTorch[4] [5], for differing values of $t$ and run linear regression analysis over our data points. From here we can then assign a $K$ value to each network and compare their loss of significance. At each $t$, the distance from the regression lines to the ideal line represents the values of $K_t$, as described in (6). From the MCDA results, AlexNet is the least sensitive to rounding and ResNet-50 is the most, with various models in between these two. Additionally we compare EfficientNet at two different model scales and evidently the larger model has much larger sensitivity. We then also compare the validation accuracy percentage decrease of all models at 10, 9 and 8-FP post-training in Figure 5. At 8-FP, besides MnasNet which experiences a large accuracy drop, $K$ is able to predict validation accuracy degradation. Thus, MCDA provides very valuable information about Q-FP model design.

## 6 CONCLUSION

We present a novel, highly sensitive, technique to quantify rounding error in DNNs. This is the first method to successfully compare the sensitivity of networks to floating point rounding error. Ultimately, this technique provides a tool for enabling the design of networks which perform higher when quantized. We do this by applying concepts from Monte Carlo Arithmetic theory to DNN computation. Furthermore, we show that by calculating the loss of significance metric $K$ from MCDA, on the CIFAR-10 and ImageNet datasets, we can compare network sensitivities to floating point rounding error and gain valuable insights to potentially design better neural networks. This is an important contribution due to the increasing interest in low-precision floating point arithmetic for efficient DNN hardware systems. The theoretical and practical contributions of this paper will likely translate well to analyzing floating point rounding in backpropagation in future work.

---

[4]https://github.com/pytorch/vision/tree/master/torchvision
[5]https://github.com/rwightman/pytorch-image-models

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

## A APPENDIX

### A.1 EXPERIMENTAL SETUP

In all our experiments, for a given network, dataset, weight representation and $t$, we run 1000 Monte Carlo trials. The network loss output from each trial is computed with the same single batch of images from the training dataset. We then independently compute the $\Theta_t$ of the network loss for $t \in \{1, 2, 3, ..., t_{max}\}$ where $t_{max} = 16$.

Following this, we then run linear regression and calculate $K$ and $t_{min}$ using MCALIB[6]. To compute the linear regression, MCALIB uses a log transformed variable, with $\log(\Theta)$ as the dependant variable and $t$ as the exploratory variable (Frechtling & Leong, 2015).

$$\log_{10}(\Theta) = \log_{10}(2^{K-t}) \tag{10}$$
$$= -\log_{10}(2)t + \log_{10}(2)K \tag{11}$$
$$= mt + c \tag{12}$$

---

[6] https://github.com/mfrechtling/mcalib

---

**Algorithm 1** Summary of Linear Regression Analysis for MCDA

---

**Initialize:** Pre-train a single-precision DNN model. Set $t_{max}$. Set number of trials $M$.
**Inputs:** Batch of inputs & targets $(X, \hat{Y})$, loss function $loss(f(x; w), y(\hat{x}))$, current weights $\boldsymbol{w}$
**Outputs:** $t_{min}$ and $K$

*Monte Carlo Trials:*
**for** $t = 1$ to $t_{max}$ **do**
    **for** trials=1 to M **do**
        $L(X; \boldsymbol{w})$ = **ForwardPath** $(X, \hat{Y}, \boldsymbol{w}, t)$ using (9)
    **end for**
    Compute $\mu$ and $\sigma$ of all trials
    Compute $\Theta_t$ using (6)
**end for**

*Calculate K and $t_{min}$:*
**for** $t = 1$ to $t_{max}$ **do**
    Compute $P_t$ using (10) - (14)
    Compute $P_t - \Theta_t$
    **if** $P_t - \Theta_t < \log_{10}(2^{0.5})$ **then**
        $t_{min} = t$
        **break**
    **else**
        **continue**
    **end if**
**end for**
Compute $K$ using (6) and (7)

---

where $m = -\log_{10}(2) = -0.30103$ is the slope and $c$ is the intercept such that $K = \log_2(10^c)$. Given these inputs, the intercept $c$ is calculated by minimizing the following objective function using Brents method (Brent, 1973) for single variable optimization:

$$f(x) = \sum_{t=1}^{t_{max}} \gamma^{t_{max}-i} \rho_H(e_i) \tag{13}$$

where $e_i = \Theta_i - (mt_i + c)$ is the residual error, $c \in [(\Theta_{t_{max}} - mt_{max}) \pm 2m]$ is the initial search space for the intercept, $\gamma = 0.75$ and $\rho_H(e)$ is the Huber loss function (Huber, 1964):

$$\rho_H(e) = \begin{cases} \frac{1}{2}e^2 & \text{for} \quad |e| \leq k \\ k\,|e| - \frac{1}{2}k^2 & \text{for } |e| > k \end{cases} \tag{14}$$

where $k = 1.345\sigma_e$ and $\sigma_e$ is the standard deviation of the residual error set, $e$. After determining the linear regression model, $P_t = mt + c$, we determine whether each $\Theta_t$ is an outlier. If a value for $\Theta_t$ differs to the equivalent $P_t$ by more than half a binary digit, then it is classed an outlier. $t_{min}$ is then defined as the lowest $t$ where $P_t - \Theta_t < \log_{10}(2^{0.5})$. To compute $K$ for a given network, we then average the values for $K_t$ whereby $t > t_{min}$. This removes the outliers from the computation of $K$. We have summarized how the experiments were simulated in Algorithm 1.

## A.2    QUANTIZED FLOATING POINT WITH STOCHASTIC ROUNDING

To quantize our pre-trained networks to Q-FP representations, we used stochastic rounding as described in (Wang et al., 2018) and implemented in QPytorch[7]. Two common forms of rounding for FP arithmetic are round-to-nearest and stochastic rounding. The former discards information in the least significant bit (LSB) which is rounded off. This information loss can be significant, especially when quantizing to a small number of bits. Stochastic rounding provides a method to capture this information loss from rounding off the LSB. Given $x = (-1)^{s_x}(1 + m_x)2^{e_x}$ as described in (1).

---

[7]https://github.com/Tiiiger/QPyTorch

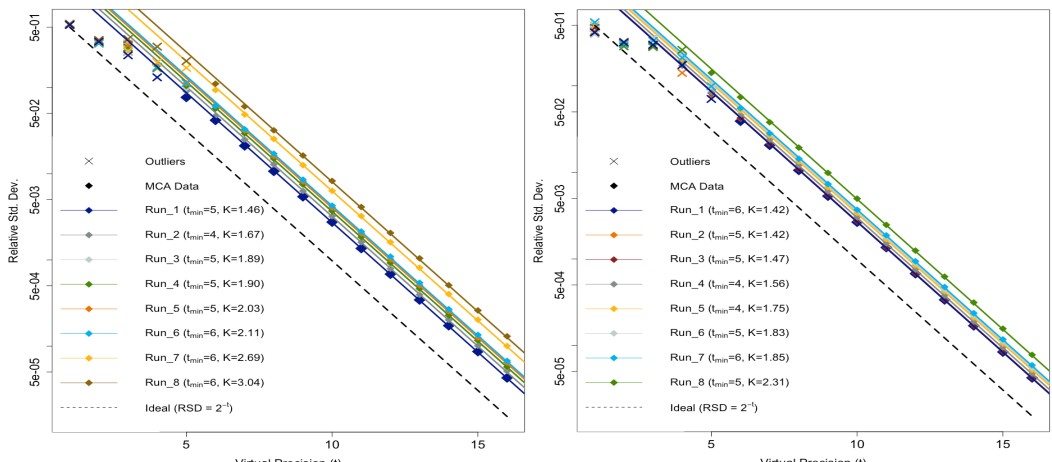

Figure 6: Comparison of $K$ and $t_{min}$ for different instances of EfficientNet-b0 (Left) and MobileNet-v2 (Right) at different virtual precisions on the ImageNet dataset.

Assume that $m_x$ is in fixed precision with $z'$ bits which needs to be rounded to $z$ bits, then stochastic rounding is as follows:

$$Round(x) = \begin{cases} (-1)^{s_x}(1 + \lfloor m_x \rfloor + \epsilon)2^{e_x} & with\,probability & \frac{m_x - \lfloor m_x \rfloor}{\epsilon} \\ (-1)^{s_x}(1 + \lfloor m_x \rfloor)2^{e_x} & with\,probability & 1 - \frac{m_x - \lfloor m_x \rfloor}{\epsilon} \end{cases} \quad (15)$$

where $\lfloor m_x \rfloor$ is the truncation of $z' - z$ LSBs of $m$, $\epsilon = 2^{-z}$. For each Q-FP accuracy reported in our experiments, we tested all possible combinations of the number of bits allowed for the exponent and mantissa which satisfied the desired precision. We then chose the combination which produced the highest accuracy.

## A.3 CIFAR-10 REGRESSION ANALYSIS

In Figure 6, we display the linear regression analysis using MCALIB for the 8 MobileNet-v2 and EfficientNet-b0 models trained on CIFAR-10. As shown, Monte Carlo trials were run for each of these models for $t \in \{1, 2, 3, ..., 16\}$. The corresponding $t_{min}$ and $K$ values were then calculated using methods discussed in Appendix A.1. Thus, in Figure 3, it is these $K$ values which are plotted against 8 and 7-bit Q-FP validation accuracy. Also, the models with lowest $K$ values are chosen for MCDA model selection in Table 1.

