# OpenReview forum: "Monte Carlo Deep Neural Network Arithmetic"
_ICLR.cc/2020/Conference — Reject_

### Official Review · AnonReviewer3 · 2019-10-20
**Official Blind Review #3**

**Rating:** 3

**Review:**

Summary:

The paper studies the sensitivity of a neural network with respect to quantizing its weights and activations. The idea is to use Monte Carlo Arithmetic (MCA) in order to calculate the number of significant bits in the training loss (e.g. cross entropy) that are lost due to floating-point arithmetic. The results show that the number of significant bits lost correlates with the reduction in classification accuracy when quantizing the weights and activations of the neural network.

Decision:

Overall, this is an interesting paper with interesting results. However, I think there is considerable room for improvement, and that more details are needed in order to assess the significance of the results, as I detail in the rest of my review. For these reasons, I recommend weak reject for now, but I encourage the authors to continue working on improving the paper and to provide more details in the updated version.

Contribution:

The paper considers an important problem, that of quantizing the weights and activations of a neural network in order to reduce computational and memory cost, while maintaining the machine-learning performance as high as possible.

In my opinion, the main contribution of the paper is the experimental findings, and in particular that the sensitivity of the training loss with respect to the precision of the weights and activations correlates with the accuracy of the network. It seems to me that these results may relate to work on Bayesian neural networks, sharp vs flat minima, and minimum-description length approaches to variational inference. Work on these areas has also shown that sensitivity of the training loss with respect to the precision of the weights (which intuitively happens when the network is at a "sharp" local minimum vs a "flat" one) is related to poor generalization performance, and vice versa. I would encourage the authors to explore the potential relationship of their work with these areas, and possibly discuss them in an updated version of the paper.

Originality:

The paper describes a method for assessing the sensitivity of a neural network with respect to the precision of the weights and activations. The method is a straightforward application of Monte Carlo Arithmetic (MCA) to neural networks. I believe that the application of MCA to neural networks for this particular purpose is novel, and that the results are original. However, the introduction of the paper gives the impression that the proposed method is brand new, and even uses the acronym MCA to refer to the proposed method, which can be confusing to readers. I would suggest to the authors to rewrite the introduction so as to reflect more accurately that the contribution is not a brand-new method, but rather the application of an existing method (MCA) in a novel way.

Writing quality:

The paper is generally easy to read, but there is considerable room for improvement. There are mistakes, and often the writing is sloppy and imprecise. I give some more specific suggestions on what to improve later on.

Technical quality:

The method is well motivated and the experiments seem reasonable. However, there is very little detail on the experiments, which makes it hard to assess their correctness/significance. I would suggest to the authors to rewrite the experimental section with full detail, or put more details in an appendix. In particular:
- Is each Monte Carlo trial done on the same batch of training images or a different one? If different, how are the trials averaged, and does that mean that the standard deviation over trials also includes a contribution due to different batches?
- In sections 5.1 and 5.2, how were the results for different t combined/aggregated? Did you use linear-regression analysis as in section 5.3?
- When you say "accuracy", do you mean accuracy on the training set, validation set, or test set? This is particularly important for assessing the significance of the results, and is something that is currently missing from the description of the experiments.
- How was the quantization of the neural networks performed? It would be good to explain this at least on a high level, in addition to citing Wang et al., (2018).
- In section 5.2, how was the model selection for each method performed exactly? In the baseline method, was the model to be quantized selected based on validation performance before quantization or after quantization?

Specific suggestions for improvement:

The citation format, i.e. (Smith et al., (2019)), is unusual and uses unnecessarily many parentheses. Use \citep for (Smith et al., 2019), and \citet for Smith et al. (2019).

The illustration of fig. 1 is not fully convincing as a motivation for floating-point arithmetic. Even though it makes the case that Float(7, 7) is more efficient than Float(8, 8) and Float (9, 9), the comparison between Float(7, 7) and Fixed(12, 12) is hard to interpret, as we can't conclude whether the efficiency gain is due to reducing the number of bits or to switching from fixed-point to floating-point arithmetic. A more convincing illustration would compare fixed-point with floating-point arithmetic using the same number of bits.

It would be better if fig. 1 were 2D, as 3D doesn't add anything but makes it harder to compare sizes visually.

Please avoid exaggerations, such as "exquisitely sensitive" or "extremely sensitive", when "sensitive" would suffice.

Section 2 is grammatically sloppy:
- arihtmetic --> arithmetic
- Last line of page 2 seems to be missing a verb.
- this has lead --> this has led

The related-work section is too short and in many cases it doesn't explain what previous work has actually done. For example, "rounding of inexact values to their nearest FP approximation has been studied in several publications" is vague: what exactly these publication have done? This lack of detail makes it hard to assess the originality of the current paper, and how it differs from existing work.

Section 3 is often unclear with imprecise mathematical notation:
- "e is the base-2 exponent of x in binary floating point arithmetic": surely, the exponent is represented as an integer?
- (bs, be1, be2, ..., bex, bm1, bm2, ..., bmx) is sloppy, as it indicates that the indices run from 1 to x.
- Bx = sx + ex + mx  is also sloppy; what is meant here is the number of bits to represent sx, ex, mx and not the values themselves.
- F(x) = x(1 + θ), shouldn't θ be δ?
- "which is typically the cause of horrific numerical inaccuracy from numerical analysis literature", the phrase "from numerical analysis literature" doesn't make much sense here.
- In eq. (4), substituting the expression for x from eq. (1) doesn't yield the same result.
- "The number of trials is an important consideration because [...] it can produce adverse effects on results". What is meant by "adverse effects"? Do you mean that with few trials Monte Carlo doesn't give accurate results? Please be more specific.
- "we can determine the expected number of significant binary digits available from a p-digit FP system as p ≥ −log2(δ)". I'm unable to follow this statement, please explain further. Also, from applying logs to δ ≤ 2^{−p} one gets an inequality that doesn't match the one in p ≥ −log2(δ).
- "The relative error of an MCA operation is, for virtual precision t, is δ ≤ 2^-t", "is" is used twice here.
- "the expected number of significant binary digits in a t-digit MCA operations is at least t", operations --> operation. Also, shouldn't it be at most t, otherwise K becomes negative?
-  is discussed in the section --> is discussed in the next section.

Some mistakes in section 4.1:
- y = (x; w) --> y = f(x; w)
- Eq. (8) is sloppy, it uses X for both the set and its size. Use |X| or something similar for the size.
- In the caption of fig. 2, baes --> base

I'm not convinced by the second bullet point in section 4.1, that the averaging over many images used to obtain the accuracy is the reason why MCA doesn't work well. Surely, the training loss (cross entropy) is also an average over many images? To me it would seem more plausible that the main reason MCA works with training loss but not accuracy is because accuracy is discrete, whereas training loss is continuous.

Fig. 3 would be much easier to read if the axes were labelled, and if the axes had the same range (so that different plots can be compared visually).

Fig. 5 would be easier to read if the networks were sorted with respect to K.

In section 5, CIFAR-10 is sometimes written as CIFAR10.

Appendix A is empty, so it should be removed.


**Experience Assessment:**

I have read many papers in this area.

**Review Assessment: Checking Correctness Of Derivations And Theory:**

I carefully checked the derivations and theory.

**Review Assessment: Checking Correctness Of Experiments:**

I carefully checked the experiments.

**Review Assessment: Thoroughness In Paper Reading:**

I read the paper thoroughly.

---

> ### Author Response · Authors · 2019-11-12
> **Response To Reviewer 3**
>
> Thank you for your review. We have provided an updated version of the paper with the main changes highlighted to address all reviewers comments.
>
> Contribution:
>
> Deeper theory and making connections with Bayesian networks are indeed interesting lines of research but we believe that it is beyond the scope of this paper. We hope that other researchers can take our work in these and other directions.
>
> Although related, our work is complementary to work using variational Bayes or Monte Carlo methods to estimate a posterior. We have updated Section 2 with a brief review of key papers and a statement regarding the new aspects of the present work.
>
> Originality:
> We agree that our references to “MCA” are misleading to the reader in regards to where the novelty of the paper lies.  We have altered Paragraph 4 of the introduction. This allows us to distinguish between our method “MCDA” and the known method “MCA”, which we have now updated throughout the paper.
>
> Writing Quality:
> We thank you for your suggestions on what to improve, it has made the updated paper more clear and concise.
>
> Technical Quality:
>
> We have added more detail to the experiments (Section 5) on how the experiments and analysis was performed.
> In answer to your particular questions:
> -Each Monte Carlo trial is done on the same batch of images in the training set. This has now been stated explicitly in Section 5.
> -In sections 5.1 and 5.2, linear regression analysis is used just as in 5.3. Different t are combined for all our experiments by averaging all the K values when t>t_min. Calculating t_min and K was done using the MCALIB library [1]. We have now added equation (7) and a more detailed description in Section 3.3 and Section 5 to make this more clear.
>
> Previously we had only cited the MCALIB paper, we have now also cited the paper and github repository in Section 5. In the appendix we have also added the linear regression curves for all models used in Sections 5.1 and 5.2 and we have explained explicitly in detail how MCALIB calculates K and t_min. This is all in the updated version of the paper.
> -We have now explicitly stated which of train and validation accuracy we are referring to at each instance.
> -We have now added an explanation of the quantization with stochastic rounding in Appendix A.3 and we reference this in Section 5.
> -In 5.2, the model chosen as the baseline was done based on whichever achieved the highest single-precision (unquantized) validation accuracy.  We have stated this more clearly in the title of Table 1 and the text in 5.2.
>
> Specific suggestions for improvement:
>
> -The citation format has now been fixed using \citep
>
> -We have replaced the Fixed(2,8) results with Fixed(8,8) results in Figure 1 as we believe this helps provide clarity for our paper motivations. We have also modified the comments on this in paragraph 3 of the Introduction.
>
> -We have fixed the grammatical areas in Section 2 and in other parts of the paper.
>
> -We have now added more papers to the related work section and have modified some of our descriptions of previous literature to be more specific. Namely, we have cited 5 more papers in relation to Monte Carlo methods for Bayesian Neural Networks, 3 of them relating specifically to quantization/compression methods. As mentioned, we have updated Section 2 with a brief review of key papers and a statement regarding the new aspects of the present work.
>
> -Section 3:
> —We have now clarified this.
> —We have now fixed this
> —We have now fixed this
> —yes it should be 𝛿, we have now fixed this
> —we have removed the wording “from numerical analysis literature” and have reworded the sentence.
> In equation (4) is now represented as (1+m) for clarity and consistency with equation (1). When subbing equation (1) into (4), it has to be reminded that the sign bit (-1)^s is not relevant to the random variable which we have defined as U~(-0.5,0.5) and hence it is discarded. Once you remove this, subbing this in does yield the same result.
> —we have removed this sentence
> —In IEEE floating point arithmetic, operators are implemented such that the error must be bounded by 𝛿. The inequality needed to be flipped, this has now been done.
> —We have now fixed this.
> —This is correct, we have now fixed this.
> —We have now fixed this.
>
> Section 4.1
> —We have now fixed this notation error
> —We have fixed equation 8 by using |X| to represent the size. We also changed the “L” to “loss” in the right hand part of the equation and in the text so that the total network loss and loss function are distinguishable.
> —This grammatical error has now been fixed.
>
> Our explanation in the second bullet point of 4.1 was incorrect. As the reviewer points out, the reason for MCA not working well is that the output is discrete. This means that, for high values of t, Monte Carlo results across different t become indistinguishable. This has now been corrected.
>
> All relevant figures and grammatical errors have been updated to address reviewers concerns.

---

### Official Review · AnonReviewer2 · 2019-10-23
**Official Blind Review #2**

**Rating:** 3

**Review:**

The authors propose a scalable method based on Monte Carlo arithmetic for quantifying the sensitivity of trained neural networks to floating point rounding errors. They demonstrate that the loss of significance metric K estimated from the process can be used for selecting networks that are more robust to quantization, and compare popular architectures (AlexNet, ResNet etc.) for their varying sensitivities.

Strengths:
- The paper tackles an important problem of analyzing sensitivity of networks to quantization and offers a well-correlated metric that can be computed without actually running models on quantized mode
- Experiments cover a wide range of architectures in image recognition

Weaknesses:
- The proposed method in Section 4.2 appears to be a straightforward modification to MCA for NN
- Experiments only demonstrate model selection and evaluating trained networks. Can this metric be used in optimization? For example, can you optimize for lowering K (say with fixed t) during training, so you can find a well-performing weight that also is robust to quantization? 1000 random samples interleaved in training may be slow, but perhaps you can use coarse approximation. This could significantly improve the impact of the paper. Some Bayesian NN literatures may be relevant (dropout, SGLD etc).

Other Comments:
- How is the second bullet point in Section 4.1 addressed in the proposed method?
- Can you make this metric task-agnostic or input-distribution-agnostic (e.g. just based on variance in predictions over some input datasets)? (e.g. you may pick a difference loss function or different test distribution to evaluate afterwards)
- Does different t give different K? If so, what’s the K reported? (are those points on Figure 3)?

**Experience Assessment:**

I do not know much about this area.

**Review Assessment: Checking Correctness Of Derivations And Theory:**

I assessed the sensibility of the derivations and theory.

**Review Assessment: Checking Correctness Of Experiments:**

I assessed the sensibility of the experiments.

**Review Assessment: Thoroughness In Paper Reading:**

I read the paper at least twice and used my best judgement in assessing the paper.

---

> ### Author Response · Authors · 2019-11-12
> **Response To Reviewer 2**
>
> Thank you for your review. We have provided an updated version of the paper with the main changes highlighted to address all reviewers comments.
>
> Weaknesses:
> - The straightforward application of MCA to NN inference on conventional GPU/CPU machines precludes most optimizations used in GEMM, and would result in a significant performance loss (explained in Section 4.1 and 4.2). Thus a key idea of this paper is to provide a technique which demonstrates how to overcome these issues for applying MCA to NN (which we call MCDA). In doing so, we gain useful insights into the sensitivity of the NN to rounding, and demonstrate that we can rank different networks, and choose good ones among those with the same loss score.
>
> -We hope our work is a precursor to further research in areas such as end-to-end low precision NN training, informative priors for Bayesian NN, training methods for weight-insensitive inference, etc. We have added additional papers on Bayesian Neural Networks to Section 2.
>
> Other Comments:
> -As pointed out by reviewer 3, a better explanation of the issue in the second bullet point of 4.1 is that the output was a discrete value. To overcome this, we use the loss function as the instead, which is a continuous value. Thus for high values of t (small perturbations), it will still present an observable change. We have explicitly stated this by modifying the second bullet point in 4.1 and the first paragraph of 4.2. in the updated version.
>
> -The statistical methods used in MCALIB (the MCA library used for our regression analysis) assume that results are normally distributed to provide the summary statistics from which K and t_min are computed. Within MCALIB, a normality test is applied, and failing indicates that changes in rounding lead to unusual changes in the (scalar) output and t_min is strictly larger than the precision for which this occurs. No assumptions regarding the distribution of the inputs nor anything about the computational graph or loss metric are made.
>
> -Different t does give different K. We now use K_t to describe K for each different t in equation (6) to make this more clear. The K for a given network is reported throughout our experiment section as the average of K values for t > t_min. We have now added equation (7) to make this more clear.
>
> For each model in Figure 3, using MCDA, we ran 1000 trials for all t in {1,2,3…,16} and calculated the relative standard deviation (RSD). We have now explicitly stated this detail, in the experiments (Beginning of Section 5). We have added to the Appendix, all the linear regression plots for all the models in Figure 3 which demonstrate this analytically. Additionally we have added further explanations and equations to the Appendix regarding how MCALIB calculates t_min and K in the regression analysis.
>
> [1] Michael Frechtling and Philip H. W. Leong.  Mcalib:  Measuring sensitivity to rounding error with monte carlo programming. ACM Trans. Program. Lang. Syst., 37(2):5:1–5:25, April 2015. ISSN 0164-0925. doi:10.1145/2665073. URL http://doi.acm.org/10.1145/2665073

---

### Official Review · AnonReviewer1 · 2019-10-23
**Official Blind Review #1**

**Rating:** 6

**Review:**

The premise of this paper is that quantization plays an important role in the deployment of deep neural networks; ie in the inference stage. However, errors due to quantization affect different neural architectures differently. It would be useful if we could predict ahead of time which models are more amenable to quantization. I think this is a very interesting premise and the paper is very well motivated.

The paper is also very clear and well written, making the claims precise and backing these up with experiments.

At the heart of the paper is the replacement of floating point numbers with inexact values, which are treated as random variables and defined precisely in equation 4. This definition enables the authors to apply Monte Carlo methods to obtain network predictions as shown in equation (10) and figure 2, and subsequently carry out sensitivity analysis. The experiments show that a measure of sensitivity (K) is indeed a good augmentation to cross-validation for model selection for the purpose of trading-off accuracy and resource consumption when launching deep neural networks with floating point rounding errors.

One question I have for the authors is the following: There has been a large body of literature on Monte Carlo methods for Bayesian neural networks. Could those works have something to say in addressing some of the challenges posed in Section 4.1?


**Experience Assessment:**

I do not know much about this area.

**Review Assessment: Checking Correctness Of Derivations And Theory:**

I assessed the sensibility of the derivations and theory.

**Review Assessment: Checking Correctness Of Experiments:**

I assessed the sensibility of the experiments.

**Review Assessment: Thoroughness In Paper Reading:**

I read the paper at least twice and used my best judgement in assessing the paper.

---

> ### Author Response · Authors · 2019-11-12
> **Response To Reviewer 1**
>
> Thank you for your review. We have provided an updated version of the paper with the main changes highlighted to address all reviewers comments.
>
> The idea of combining MCA with Bayesian Neural Networks is indeed interesting, but beyond the scope of this paper. We hope that this paper will provide an initial direction for further research in MCA for NNs which enables deeper understanding of quantization in inference and training. In response to your comments and those of the other reviewers, we have cited an additional 5 papers concerning Bayesian Neural Networks in Section 2.

---

### Author Response · Authors · 2019-11-15
**Summary Of Changes Made**

We thank all the reviewers for the constructive comments and valuable suggestions. We have uploaded a revised version of our paper following the suggestions. In the revised paper, we have highlighted the main changes in blue/aqua. The changes for each section can be summarized as follows:

In Section 1, we updated Figure 1 to show a more insightful operator comparison. We now directly compare fixed and floating point of the same precision width of 8bits. This more clearly shows that a drop of a single bit of precision in floating point, from 8->7, can have significant hardware benefits and shows improvement over the 8-bit fixed point operator. We also make a clear distinguishment between our work 'MCDA' and previous work 'MCA'.

In section 2, we further discuss the literature and contrast our work to others on Bayesian NNs.

In section 3, we fix some of the notation and wording for correctness and add an equation to more clearly describe how we calculated the loss of significance value, K.

In section 4, we fix some of the notation and the wording for the issues of applying traditional MCA to DNNs. We also remove an equation which we deemed unnecessary.

In section 5, we provide much greater detail on how we ran MCDA experiments, including a reference to the MCALIB repository which we used to produce regression plots for calculating K and t_min. We also fix the scale and labelling of the axes in Figure 3 and we order the networks in Figure 5 based on their respective K values. This makes the figures more clear and coherent.

We also added an appendix which describes how MCALIB calculates k and t_min specifically. Additionally, we add more detail on the quantization function used to produce our quantized floating point representations.

---

### Decision · Program_Chairs · 2019-12-19

**Decision:**

Reject

**Comment:**

The paper studies the impact of rounding errors on deep neural networks. The
authors apply Monte Carlos arithmetics to standard DNN operations.
Their results indeed show catastrophic cancellation in DNNs and that the resulting loss of
significance in the number representation correlates with decrease in validation
performance, indicating that DNN performances are sensitive to rounding errors.

Although recognizing that the paper addresses an important problem (quantized /
finite precision neural networks), the reviewers point out the contribution of
the paper is somewhat incremental.
During the rebuttal, the authors made an effort to improve the manuscript based
on reviewer suggestions, however review scores were not increased.

The paper is slightly below acceptance threshold, based on reviews and my own
reading, as the method is mostly restricted to diagnostics and cannot yet be used
to help training low-precision neural networks.